Problems in using p-curve analysis and text-mining to detect rate of p-hacking and evidential value

Bishop Dorothy V.M. dorothy.bishop@psy.ox.ac.uk
Thompson Paul A.
Department of Experimental Psychology, University of Oxford , Oxford , United Kingdom
Chen Jun
Electronic publication date: 2016 Feb 18
Publication date: 2016
Volume: 4
Electronic Location ID: e1715
Received 2015 Sep 1; Accepted 2016 Jan 29
Copyright: ©2016 Bishop and Thompson
Copyright year: 2016
Copyright holder: Bishop and Thompson
License: This is an open access article distributed under the terms of the Creative Commons Attribution License, which permits unrestricted use, distribution, reproduction and adaptation in any medium and for any purpose provided that it is properly attributed. For attribution, the original author(s), title, publication source (PeerJ) and either DOI or URL of the article must be cited.
License URL: https://creativecommons.org/licenses/by/4.0/

Keywords: Reproducibility, p-hacking, Simulation, Ghost variables, Text-mining, p-curve, Correlation, Power

Funding: Wellcome Trust Principal Research Fellowship Wellcome Trust Programme 082498/Z/07/Z The first author is supported by a Wellcome Trust Principal Research Fellowship. Both authors are funded by Wellcome Trust Programme Grant number 082498/Z/07/Z. The funders had no role in study design, data collection and analysis, decision to publish, or preparation of the manuscript.

==============================
Background. The p-curve is a plot of the distribution of p-values reported in a set of scientific studies. Comparisons between ranges of p-values have been used to evaluate fields of research in terms of the extent to which studies have genuine evidential value, and the extent to which they suffer from bias in the selection of variables and analyses for publication, p-hacking.

Methods. p-hacking can take various forms. Here we used R code to simulate the use of ghost variables, where an experimenter gathers data on several dependent variables but reports only those with statistically significant effects. We also examined a text-mined dataset used by Head et al. (2015) and assessed its suitability for investigating p-hacking.

Results. We show that when there is ghost p-hacking, the shape of the p-curve depends on whether dependent variables are intercorrelated. For uncorrelated variables, simulated p-hacked data do not give the “p-hacking bump” just below .05 that is regarded as evidence of p-hacking, though there is a negative skew when simulated variables are inter-correlated. The way p-curves vary according to features of underlying data poses problems when automated text mining is used to detect p-values in heterogeneous sets of published papers.

Conclusions. The absence of a bump in the p-curve is not indicative of lack of p-hacking. Furthermore, while studies with evidential value will usually generate a right-skewed p-curve, we cannot treat a right-skewed p-curve as an indicator of the extent of evidential value, unless we have a model specific to the type of p-values entered into the analysis. We conclude that it is not feasible to use the p-curve to estimate the extent of p-hacking and evidential value unless there is considerable control over the type of data entered into the analysis. In particular, p-hacking with ghost variables is likely to be missed.

Background

Statistical packages allow scientists to conduct complex analyses that would have been impossible before the development of fast computers. However, understanding of the conceptual foundations of statistics has not always kept pace with software (Altman, 1991; Reinhart, 2015), leading to concerns that much reported science is not reproducible, in the sense that a result found in one dataset is not obtained when tested in a new dataset (Ioannidis, 2005). The causes of this situation are complex and the solutions are likely to require changes, both in training of scientists in methods and revision of the incentive structure of science (Ioannidis, 2014; Academy of Medical Sciences et al., 2015).

Two situations where reported p-values provide a distorted estimate of strength of evidence against the null hypothesis are publication bias and p-hacking. Both can arise when scientists are reluctant to write up and submit unexciting results for publication, or when journal editors are biased against such papers. Publication bias occurs when a paper reporting positive results—e.g., those that report a significant difference between two groups, an association between variables, or a well-fitting model of a dataset—are more likely to be published than null results (Ioannidis et al., 2014). Concerns about publication bias are not new (Greenwald, 1975; Newcombe, 1987; Begg & Berlin, 1988), but scientists have been slow to adopt recommended solutions such as pre-registration of protocols and analyses.

The second phenomenon, p-hacking, is the focus of the current paper. It has much in common with publication bias, but whereas publication bias affects which studies get published, p-hacking is a bias affecting which data and/or analyses are included in a publication arising from a single study. p-hacking has also been known about for many years; it was described, though not given that name, in 1956 (De Groot, 2014). The term p-hacking was introduced by Simonsohn, Nelson & Simmons (2014) to describe the practice of reporting only that part of a dataset that yields significant results, making the decision about which part to publish after scrutinising the data. There are various ways in which this can be done: e.g., deciding which outliers to exclude, when to stop collecting data, or whether to include covariates. Our focus here is on what we term ghost variables: dependent variables that are included in a study but then become invisible in the published paper after it is found that they do not show significant effects.

Although many researchers have been taught that multiple statistical testing will increase the rate of type I error, lack of understanding of p-values means that they may fail to appreciate how use of ghost variables is part of this problem. If we compare two groups on a single variable and there is no genuine difference between the groups in the population, then there is a one in 20 chance that we will obtain a false positive result, i.e., on a statistical test the means of the groups will differ with p < .05. If, however, the two groups are compared on ten independent variables, none of which differs in the overall population, then the probability that at least one of the measures will yield a ‘significant’ difference at p < .05 is 1 − (1 − 0.05)10, i.e., .401 (De Groot, 2014). So if a researcher does not predict in advance which measure will differ between groups, but just looks for any measure that is ‘significant,’ there is a 40% chance they will find at least one false positive. If they report data on all 10 variables, then statistically literate reviewers and editors may ask them to make some correction for multiple comparisons, such as the Bonferroni correction, which requires a more stringent significance level when multiple exploratory tests are conducted. If, however, the author decides that only the significant results are worth reporting, and assigns the remaining variables to ghost status, then the published paper will be misleading in implying that the results are far more unlikely to have occurred by chance than is actually the case, because the ghost variables are not reported. It is then more likely that the result will be irreproducible. Thus, use of ghost variables potentially presents a major problem for science because it leads to a source of irreproducibility that is hard to detect, and is not always recognised by researchers as a problem (Kraemer, 2013; Motulsky, 2015).

Simonsohn, Nelson & Simmons (2014) proposed a method for diagnosing p-hacking by considering the distribution of p-values obtained over a series of independent studies. Their focus was on the p-curve in the range below .05, i.e., the distribution of probabilities for results meeting a conventional level of statistical significance. The logic is that a test for a group difference when there is really no effect will give a uniform distribution of obtained p-values. In contrast, when there is a true effect, repeated studies will show a right-skewed p-curve, with p-values clustered at the lower end of the distribution (see Fig. 1). As shown by Simonsohn, Nelson & Simmons (2014), the degree of right skew will be proportional to sample size (N), as we have more power in the study to detect real group differences when N is large (Cohen, 1992).

Figure 1 P-curve: expected distribution of p-values when no effect (null) vs true effect size of 0.3 with low (N = 20 per group) or high power (N = 200 per group).

Simonsohn, Nelson & Simmons (2014) went on to show that under certain circumstances, p-hacking can lead to a left-skewing of the p-curve, with a rise in the proportion of p-values that are just less than .05. This can arise if researchers adopt extreme p-hacking methods, such as modifying analyses with covariates, or selectively removing subjects, to push ‘nearly’ significant results just below the .05 threshold.

Demonstrations of the properties of p-curves has led to interest in the idea that they might be useful to detect whether p-hacking is present in a body of work. Although p-curves have been analysed using curve-fitting (Masicampo & Lalande, 2012), it is possible to use a simple binomial test to detect skew near .05, characteristic of p-hacking and, conversely, to use the amount of right skew to estimate the extent to which a set of studies gives results that are likely to be reproducible, i.e., has evidential value. In a recent example, Head et al. (2015) used text-mined p-values from over 111,000 published papers in different scientific disciplines. For each of 14 subject areas, they selected one p-value per paper to create a p-curve that was then used to test two hypotheses. First, they used the binomial test to compare the number of significant p-values in a lower bin (between 0 and .025) with the number in a higher bin (between .025 and .05). As shown in Fig. 1, if there are no true effects, then we expect equal proportions of p-values in these two bins. Therefore, they concluded that if there were significantly more p-values in the lower bin than the higher bin, this was an indication of ‘evidential value’, i.e., results in that field were true findings. Next, they compared the number of p-values between two adjacent bins near the significance threshold of .05: a far bin (.04 < p < .045) and a near bin (.045 < p < .05). If there were more p-values in the near bin than the far bin, they regarded this as evidence of p-hacking.

Questions have, however, been raised as to whether p-curves provide a sufficiently robust foundation for such conclusions. Simonsohn, Nelson & Simmons (2014) emphasised the assumptions underlying p-curve analysis, and the dangers of applying the method when these were not met. Specifically, they stated, “For inferences from p-curve to be valid, studies and p-values must be appropriately selected…. selected p-values must be (1) associated with the hypothesis of interest, (2) statistically independent from other selected p-values, and (3) distributed uniform under the null” (p. 535) (i.e., following the flat function illustrated in Fig. 1). Gelman & O’Rourke (2014) queried whether the requirement for a uniform distribution was realistic. They stated: “We argue that this will be the case under very limited settings”, and “The uniform distribution will not be achieved for discrete outcomes (without the addition of subsequent random noise), or for instance when a t.test is performed using the default in the R software with small sample sizes (unequal variances).” (p. 2–3)

The question, therefore, arises as to how robust p-curve analysis is to violations of assumptions regarding the underlying data, and under what circumstances it can be usefully applied to real-world data. To throw light on this question we considered one factor that is common in reported papers: use of correlated dependent variables. We use simulated data to see how correlation between dependent measures affects the shape of the p-curve when ghost p-hacking is adopted (i.e., several dependent measures are measured but only a subset with notionally ‘significant’ results is reported). We show that, somewhat counterintuitively, ghost p-hacking induces a leftward skew in the p-curve when the dependent variables are intercorrelated, but not when they are independent.

Another parameter in p-curve analysis is the number of studies included in the p-curve. The study by Head et al. (2015) exemplifies a move toward using text-mining to harvest p-values for this purpose, and their study therefore was able to derive p-curves based on a large number of studies. When broken down by subject area, the number of studies in the p-curve ranged from around 100 to 62,000. It is therefore of interest to consider how much data is needed to have reasonable power to detect skew.

Finally, with text-mining of p-values from Results sections we can include large numbers of studies, but this approach introduces other kinds of problems: not only do we lack information about the distributions of dependent variables and correlations between these, but we cannot even be certain that the p-values are related to the main hypothesis of interest. We conclude our analysis with scrutiny of a subset of studies used by Head et al. (2015), showing that their analysis included p-values that were not suitable for p-curve analysis, making it unfeasible to use the p-curve to quantify the extent of p-hacking or evidential value.

Materials and Methods

Simulations

A script, Ghostphack, was written in R to simulate data and derive p-curves for the situation when a researcher compares two groups on a set of variables but then reports just those with significant effects. We restrict consideration to the p-curve in the range from 0 to .05. Ghostphack gives flexibility to vary the number of variables included, the effect size, the inter-correlation between variables, the sample size, the extent to which variables are normally distributed, and whether or not p-hacking is used. p-hacking is simulated by a model where the experimenter tests X variables but only reports the subset that have p < .05; both one-tailed (directional) and two-tailed versions can be tested.

As illustrated in Appendix S1, each run simulates one study in which a set of X variables is measured for N subjects in each of two groups. In each run, a set of random normal deviates is generated corresponding to a set of dependent variables. In the example, we generate 40 random normal deviates, which correspond to four dependent variables measured on five participants in each of two groups, A and B. The first block of five participants is assigned to group A and the second block to group B. If we are simulating the situation where there is a genuine difference between groups on one variable, an effect size, E, is added to one of the dependent variables for group A only. A t-test is then conducted for each variable to test the difference in means between groups, to identify variables with p < .05. In practice, there may be more than one significant p-value per study, and we would expect that researchers would report all of these; however, for p-curve analysis, it is a requirement that p-values are independent (Simonsohn, Nelson & Simmons, 2014), and so only one significant p-value is selected at random per study for inclusion in the analysis. The analysis discards any studies with no significant p-values. The script yields tables that contain information similar to that reported by Head et al. (2015): the number of runs with p-values in specific frequency bins.

All simulations reported here were based on 100,000 runs, each of which simulated a study with either 3 or 8 dependent variables for two groups of subjects. Two power levels were compared: low (total N of 40, i.e., 20 per group) and high (total N of 400, i.e., 200 per group).

Effect of correlated data on the p-value distribution

In the example in Appendix S1, the simulated variables are uncorrelated. In practice, however, studies are likely to include several variables that show some degree of intercorrelation (Meehl, 1990). We therefore compared p-curves based on situations where the dependent variables had different degrees of intercorrelation. We considered situations where researchers measure multiple response variables that are totally uncorrelated, weakly correlated, or strongly correlated with each other, and then only report one of the significant ones.

An evaluation of text-mined p-curves

Text-mining of published papers makes it possible to obtain large numbers of studies for p-curve analysis. In the final section of this paper, we note some problems for this approach, illustrated with data from Head et al. (2015).

Results

Simulations: correlated vs. uncorrelated variables

Figure 2 shows output from Ghostphack for low (N = 20 per group) and high (N = 200 per group) powered studies when data are sampled from a population with no group difference. Figures 2A and 2B show the situation when there are 3 variables, and Figures 2C and 2D with 8 variables. Intercorrelation between the simulated variables was set at 0, .5, or .8. Directional t-tests were used; i.e., a variable was treated as a ghost variable only if there was a difference in the predicted direction, with greater mean for group 2 than for group 1.

Figure 2 P-curve for ghost p-hacked data when true effect size is zero (A and C) versus when true effect is 0.3 (B and D).

Continuous line for low power (N = 20 per group) and dashed line for high power (N = 200 per group). Different levels of correlation between variables are colour coded.

For uncorrelated variables, using data generated with a null effect, the p-hacked p-curve is flat, whereas for correlated variables, it has a negative skew, with the amount of slope a function of the strength of correlation. The false positive rate is around 40% when variables are uncorrelated, but drops to around 12% when variables are intercorrelated at r = .8. Figure 2 also shows how the false positive rate increases when the number of variables is large (8 variables vs. 3 variables)—this is simply a consequence of the well-known inflation of false positives when there are multiple comparisons.

The slope of the p-curve with correlated variables is counterintuitive, because if we plot all obtained p-values from a set of t-tests when there is no true effect, this follows a uniform distribution, regardless of the degree of correlation. The key to understanding the skew is to recognise it arises when we sample one p-value per paper. When variables are intercorrelated, so too are effect sizes and p-values associated with those variables. It follows that for any one run of Ghostphack, the range of obtained p-values is smaller for correlated than uncorrelated variables, as shown in Table 1. In the limiting case where variables are multicollinear, they may be regarded as indicators of a single underlying factor, represented by the median p-value of that run. Across all runs of the simulation, the distribution of these median values will be uniform. However, sampling according to a cutoff from correlated p-values will distort the resulting distribution: if the median p-value for a run is well below .05, as in the 2nd row of Table 1B, then most or all p-values from that run will be eligible. However, if the median p-value is just above .05, as in the final row of Table 1B, then only values close to the .05 boundary are eligible for selection. In contrast, when variables are uncorrelated, there are no constraints on any p-values, and all values below .05 are equally likely. See also comment by De Winter and Van Assen on Bishop & Thompson (2015: version 2 of this paper), which elaborates on this point.

Table 1 Rank-ordered p-values for 10 runs of simulation with (A) r = 0, and (B) r = .8.

Values less than .05 which are candidates for inclusion in p-curve are shown with pink highlight.

p1	p2	p3	p4	p5	p6	p7	p8	p9	p10	Median p	Range	
(A) Correlation between variables =0	
0.030	0.208	0.259	0.564	0.715	0.807	0.832	0.875	0.895	0.969	0.761	0.939	
0.049	0.050	0.276	0.332	0.472	0.479	0.785	0.804	0.936	0.974	0.475	0.925	
0.085	0.164	0.383	0.456	0.470	0.481	0.600	0.615	0.718	0.839	0.476	0.754	
0.006	0.181	0.202	0.244	0.315	0.325	0.359	0.443	0.471	0.635	0.320	0.629	
0.332	0.351	0.411	0.426	0.505	0.611	0.648	0.713	0.884	0.913	0.558	0.581	
0.076	0.160	0.266	0.276	0.309	0.328	0.342	0.346	0.422	0.964	0.319	0.888	
0.046	0.053	0.105	0.227	0.508	0.508	0.800	0.819	0.885	0.973	0.508	0.927	
0.048	0.101	0.234	0.264	0.414	0.433	0.606	0.709	0.788	0.968	0.424	0.921	
0.051	0.113	0.282	0.445	0.452	0.456	0.656	0.670	0.736	0.757	0.454	0.705	
0.082	0.202	0.221	0.241	0.297	0.383	0.387	0.717	0.955	0.982	0.340	0.900	
(B) Correlation between variables =.8	
0.110	0.172	0.375	0.449	0.508	0.575	0.633	0.644	0.747	0.787	0.541	0.677	
0.001	0.004	0.006	0.007	0.007	0.010	0.012	0.013	0.043	0.060	0.009	0.059	
0.602	0.775	0.820	0.853	0.859	0.889	0.933	0.942	0.950	0.956	0.874	0.353	
0.128	0.211	0.227	0.229	0.252	0.255	0.342	0.368	0.450	0.571	0.253	0.443	
0.218	0.249	0.328	0.338	0.392	0.489	0.557	0.561	0.604	0.877	0.441	0.660	
0.519	0.801	0.848	0.893	0.903	0.939	0.948	0.984	0.990	0.997	0.921	0.477	
0.179	0.260	0.331	0.344	0.385	0.425	0.455	0.608	0.758	0.765	0.405	0.585	
0.569	0.575	0.627	0.639	0.746	0.749	0.780	0.901	0.906	0.920	0.747	0.351	
0.210	0.284	0.379	0.418	0.474	0.570	0.593	0.654	0.670	0.790	0.522	0.580	
0.013	0.084	0.091	0.099	0.121	0.154	0.156	0.36	0.435	0.439	0.137	0.426	

Figures 2B and 2D also shows the situation where there is a true but modest effect (d = .3) for one variable. Here we obtain the signature right-skewed p-curve, with the extent of skew dependent on the statistical power, but little effect of the number of dependent variables. Appendix S2 shows analogous p-curves for plots simulated with the same parameters and no p-hacking: the p-curve is flat for the null effect; for the effect of 0.3, a similar degree of right-skewing is seen as in Fig. 2, but in neither case is there any influence of correlation between variables (see Appendix S2). For completeness, Appendix S2 also shows p-curves with the y-axis expressed as percentage of p-values, rather than counts.

In real world applications we would expect p-values entered into a p-curve to come from studies with a mixture of true and null effects, and this will affect the ability to detect the right skew indicative of evidential value, as well as the left skew. Lakens (2014) noted that a right-skewed p-curve can be obtained even when the proportion of p-hacking is relatively high. Nevertheless, the left-skewing caused by correlated variables complicates the situation, because when power is low and we have highly correlated variables, inclusion of a proportion of p-hacked trials can cancel out the right skew because of the left skew induced by p-hacking with correlated variables (see Fig. 3). This is just one way in which the combination of parameters can yield unexpected effects on a p-curve: this illustrates the difficulty of interpreting p-curves in real-life situations where parameters such as proportion of p-hacked studies, sample size and number and correlation of dependent variables are not known. Such cases appear to contradict the general rule of Simonsohn, Nelson & Simmons (2014) that: “all combinations of studies for which at least some effects exist are expected to produce right-skewed p-curves.” (p. 536), because the right skew can be masked if the set of p-values includes a subset from low-powered null studies that were p-hacked from correlated ghost variables. Our point is that interpretation of the p-curve must not be taken as definitive evidence of the presence or lack of p-hacking or evidential value, although it can indicate that something is problematic.

Figure 3 Illustration of how right skew showing evidential value can be masked if there is a high proportion of p-hacked studies and low statistical power.

Colours show N, and continuous line is non-hacked, dotted line is p-hacked.

Power to detect departures from uniformity in the range p = 0–.05

We have noted how power of individual studies will affect p-curves, but there is another aspect of power that also needs to be considered, namely the power of the p-curve analysis itself. We restrict consideration here to the simple method adopted by Head et al. (2015), where the number of p-values is compared across two ranges. For instance, to detect the ‘bump’ in the p-curve just below .05, we can compare the number of p-values in the bins .04 <p < .045 (far) vs. .045 < p < .05 (near). These numbers will depend on (a) the number of studies included in the p-curve analysis; (b) the proportion of studies where ghost p-hacking was used; (c) the number of variables in the study; (d) the sample size and (e) the correlation between variables. Consider an extreme case, where we have no studies with a true effect, with ghost-hacking in all studies, and eight variables with inter-correlation of .8. This set of parameters leads to slight left-skewing of the p-curve (Fig. 3). Simulated data were used to estimate the proportions of p-values in the near and far bins close to .05, and hence to derive the statistical power to detect such a difference. To achieve 80% power to detect a difference, a total of around 1,200 p-values in the range between .04 and .05 is needed. Note that to find this many p-values, considerably more studies would be required. In the simulation used for Fig. 4, only 4% of simulated studies had p-values that fell in this range. It follows that to detect the p-hacking bump with 80% power in this situation, where the difference due to ghost p-hacking is maximal, we would need p-values from 30,000 studies.

Figure 4 Power curve for detecting difference between near and far p-value bins in case with null effect, 100% ghost p-hacking, and eight variables with intercorrelation of 0.8.

N.B. the saw-tooth pattern is typical for this kind of power curve (Chernick & Liu, 2002).

Text-mined p-curves

For their paper entitled “The extent and consequences of P-hacking in science,” Head et al. (2015) downloaded all available open access papers from PubMed Commons, categorised them by subject area, and used text-mining to locate Abstracts and Results sections, and then to search in these for p-values in the range 0–.05. One p-value was randomly sampled per paper. This sampling was repeated 1,000 times, and the rounded average number of p-values in a given bin was taken as the value used in the p-curve for that paper. The number of papers included by Head et al. (2015) varied considerably from discipline to discipline, from 94 in Mathematical Sciences to over 60,000 for Medical and Health Sciences. These were divided according to whether p-values came from Results or Abstracts sections. This is, to our knowledge, the largest study of p-hacking in the literature.

Although this approach to p-hacking has the merit of using massive amounts of data, problems arise from the lack of control over p-values entered into the analysis.

Ambiguous p-values in text-mined data

Some reported p-values are inherently ambiguous. In their analysis of text-mined data, Head et al. (2015) included p-values in the p-curve only if they were specified precisely (i.e., using ‘=’). Use of a ‘less than’ specifier was common for very low values, e.g., p < .001, but these were omitted. We manually checked a random subset of 30 of the 1,736 papers in the Head et al. dataset classified as Psychology and Cognitive Sciences (see Appendix S3 for DOIs). The average number of significant p-values reported in each paper was 14.1, with a range from 2 to 43. If values specified as <.01 or <.001 were included in the bin ranging from 0 to .025, then for the 30 papers inspected in detail, the average number per paper was 9.97; if they were excluded (as was done in the analysis by Head et al.) then the average number was 4.47, suggesting that around half the extreme p-values were excluded from analysis because they were specified as ‘less than’, even though they could accurately have been assigned to the lowest bin. However, if all these values had been included in the analysis, then Head et al. might have been accused of being biased in favour of finding extreme p-values; in this regard, the approach they adopted was very conservative, reducing the power of the test. Another problem is variability in the number of decimal places used to report p-values, e.g., if we see p = .04, it is unclear if this is a precise estimate or if it has been rounded. Head et al. (2015) dealt with this issue by including only p-values reported to at least three decimal places, but alternative solutions to the problem will give different distributions of p-values.

Unsuitable p-values in text-mined data

As Simonsohn, Nelson & Simmons (2014) and Simonsohn, Simmons & Nelson (2015) noted, it is important to select carefully the p-values for inclusion in a p-curve. Scrutiny of the 30 papers from Head et al. (2015) selected for detailed analysis (Appendix S3) raised a number of issues about the accuracy of p-curve analysis of text-mined data:

1. Perhaps the most serious issue concerns cases where p-values extracted from the mined text could exaggerate evidential value. There were numerous instances where p-values were reported that related to facts that were either well-established in the literature, or strongly expected a priori, but which were not the focus of the main hypothesis; the impression was that these were often reported for completeness and to give reassurance that the data conformed to general expectations. For instance, in paper 1, a very low p-value was found for the association between depression and suicidality—not a central focus of the paper, and not a surprising result. In paper 10, which looked at the effect of music on verbal learning, a learning effect was found with p < .001—this simply demonstrated that the task used by the researchers was valid for measuring learning. This strong effect affected several p-values because it was further tested for linear and quadratic trends, both of which were significant (with p = .004 and p < .001). None of these p-values concerned a test of the primary hypothesis. In paper 11, a statistical test was done to confirm that negative photos elicited more negative emotion than positive photos—and gave p < .001; again, this was part of an analysis to confirm the suitability of the materials but it was not part of the main hypothesis-testing. Study 20, on Stroop effects in bilinguals, reported a highly significant Stroop effect, an effect so strong and well-established that there is little interest in demonstrating it beyond showing the methods were sound. Examples such as these could be found in virtually all the papers examined.

2. Some papers had evidence of double-dipping (Kriegeskorte et al., 2009), a circular procedure commonly seen in human brain mapping, when a large dataset is first scrutinised to identify a region that appears to respond to a stimulus, and then analysis is focused on that region. This is a practice that is commonplace in electrophysiological as well as brain imaging studies; For instance, in paper 6, an event-related potentials study, a time range where two conditions differed was first identified by inspection of average waveforms, and then mean amplitudes in this interval were compared across conditions. This is a form of p-hacking that can generate p-values well below .05. For instance, Vul et al. (2009) showed that where such circular analysis methods had been used, reported correlations between brain activation and behaviour often exceeded 0.74. Even with the small sample sizes that are often seen in this field, this would be highly significant (e.g., for N = 16, p < .001). This would not be detected by looking for a bump just below .05, but rather would give the false impression of evidential value.

3. For completeness we note also two other cases where p-values would not be suitable for p-curve analysis, (a) where they are associated with tests of assumptions of a method and (b) in model-fitting contexts, where a low p-value indicates poor model fit. Examples of these were, however, rare in the papers we analysed; two papers reported values for Mauchly’s tests of sphericity of variances, but only one of these was reported exactly, as p = .001, and no study included statistics associated with model-fitting. So although such cases could give misleading indications of evidential value, they are unlikely to affect the p-curve except in sub-fields where use of such statistics is common.

Discussion

Problems specific to text-mined data

Automated text-mining provides a powerful means for extracting statistics from very large databases of published texts, but the increased power that this provides comes at a price, because the method cannot identify which p-values are suitable for inclusion in p-curve analysis. Simonsohn, Nelson & Simmons (2014) and Simonsohn, Simmons & Nelson (2015) argued that p-curve analysis should be conducted on p-values that meet three criteria: they test the hypothesis of interest, they have a uniform distribution under the null, and they are statistically independent of other p-values in the p-curve. The text-mined data from Results section used by Head et al. (2015) do not adhere to the first requirement. Most scientific papers include numerous statistical tests, only some of which are specifically testing the hypothesis of interest. If one simply assembles all the p-values in a paper and selects one at random, this avoids problems of dependence between p-values, but it means that unsuitable p-values will be included. Table 2 summarises the problems that arise when p-curve analysis is used to detect p-hacking and evidential value from text-mined data.

Table 2 Problems in quantifying p-hacking and evidential value from a p-curve using text-mined data.

Cases where p-hacking not detected by binomial test	Cases where right skew not due to evidential value	
P-values are reported as p < .05 and so excluded from analysisa	Where p-values used to confirm prior characteristics of groups being compareda,b	
Limited power because few p-values between .04 and .05	Where p-values come from confirming well-known effects, e.g., demonstrating that a method behaves as expecteda,b	
Where p-values ambiguous because rounded to two decimal placesa	Where ‘double-dipping’ used to find ‘best’ data to analyse	
	P-values from model-fitting or testing of assumptions of statistical tests (where low p-value indicative of poor fit, or failure to meet assumptions)a,b	
Notes.

a Problems that can potentially be overcome by analysing data from meta-analyses.

b Problems that are less likely to affect text-mined data from Abstracts.

Most of these problems are less likely to affect text-mined data culled from Abstracts. As De Winter & Dodou (2015) noted, p-values reported in Abstracts are likely to be selected as relating to the most important findings. Indeed, studies that have used text-mining to investigate the related topic of publication bias have focused on Abstracts, presumably for this reason, e.g., Jager & Leek (2013) and De Winter & Dodou (2015). However, reporting of p-values in Abstracts is optional and many studies do not do this; there is potential for bias if the decision to report p-values in the Abstract depends on the size of the p-value. Furthermore, it is difficult to achieve adequate statistical power to test for the p-hacking bump. With their extremely large set of Abstracts, Head et al. (2015) found evidence of p-hacking in only two of the ten subject areas they investigated, but in six areas there were less than ten p-values between .04 and .05 to be entered into the analysis.

As noted in Table 2, many of these problems can be avoided by using meta-analyses, where p-values have been selected to focus on those that tested specific hypotheses. Head et al. (2015) included such an analysis in their paper, precisely for this reason. However, such an analysis is labour-intensive, and has limited power to detect p-hacking if the overall number of p-values in the .04–.05 range is small (see Head et al., 2015, Table 3)

More general problems with drawing inferences from binomial tests on p-curves

Lakens (2015) noted that to model the distribution of p-values we need to know the number of studies where the null hypothesis or alternative hypothesis is true, the nominal type I error rate, the statistical power and extent of publication bias. We would add that we also need to know whether dependent variables were correlated, whether p-values were testing a specific hypothesis, and how many p-values had to be excluded (e.g., because of ambiguous reporting).

Our simulations raise concerns about drawing conclusions from both ends of the p-curve. In particular, we argue that the binomial test cannot be used to quantify the amount of p-hacking. These interpretive problems potentially apply to all p-curves, not just those from text-mined data.

As we have shown, one form of p-hacking, ghost p-hacking, does not usually lead to a significant difference between the adjacent bins close to the .05 cutoff. In particular, where there is ghost p-hacking with variables that are uncorrelated or weakly correlated the p-curve is flat across its range. Where ghost p-hacked variables are correlated, a leftward skew is induced, which increases with the degree of correlation, but our power analysis showed that very large numbers of studies would need to be entered into a p-curve for this to be detected. In such cases, a binomial test of differences between near and far bins close to .05 will give a conservative estimate of p-hacking. Use of ghost variables is just one method of p-hacking, and the ‘bump’ in the p-curves observed by Head et al. could have resulted for other reasons: indeed, in an analysis of meta-analysed studies, they showed that a contributing factor was authors misreporting p-values as significant (when recomputation showed they were actually greater than .05). Our general point, however, is that without more information about the data underlying a p-curve, it can be difficult to interpret the absence of a p-hacking ‘bump’. In fact, virtually all meta-analytic techniques, e.g., trim and fill, that try to correct for bias are subject to certain assumptions, and when these are not adhered to, this creates difficulty in interpretation of results.

Right skewing provides evidential value, but with heterogeneous data it is difficult to quantify the extent of this from the degree of rightward skew in a p-curve, because, as already noted by Simonsohn, Nelson & Simmons (2014), this is dependent on statistical power. In particular, as we have shown, when a dataset contains ghost p-hacked correlated variables, these have little impact when the statistical power is high, but can counteract the right skewing completely when power is low.

We share the concerns of Head et al. (2015) about the damaging impact of p-hacking on science. On the basis of p-curve analysis of meta-analysed data, they concluded that “while p-hacking is probably common, its effect seems to be weak relative to the real effect sizes being measured.” (p. 1). As we have shown here, if we rely on a ‘bump’ below the .05 level to detect p-hacking, it is likely that we will miss much p-hacking that goes ‘under the radar’. P-curve analysis still has a place in contexts where probabilities are compared for a set of p-values (pp-values) from a series of studies that are testing a hypothesis, and which meet the criteria of Simonsohn, Nelson & Simmons (2014) and Simonsohn, Simmons & Nelson (2015). However, simple comparisons between ranges of p-values in data from disparate studies do not allow us to quantify the extent of either p-hacking or real effects.

Supplemental Information

Appendix S1 Appendices 1–3

Click here for additional data file.

We are most grateful to Head et al. (2015) for making scripts and data publicly available, and for engaging in discussion about the points raised in a preprint of this paper, and specifically for providing the script by Luke Holman, which provides a useful alternative method for simulating ghost p-hacking. A slightly modified version of this script, which we used to generate some plots, is now available with our other scripts. We thank also Joost de Winter and Daniel Lakens for their contributions in helping us develop this paper.

Additional Information and Declarations

Competing Interests

Author Contributions

Data Availability

Dorothy V. Bishop is an Academic Advisor and an Academic Editor for PeerJ.

Dorothy V. Bishop conceived and designed the experiments, analyzed the data, wrote the paper, prepared figures and/or tables, reviewed drafts of the paper, programming simulation.

Paul A. Thompson analyzed the data, prepared figures and/or tables, reviewed drafts of the paper, programming simulation.

The following information was supplied regarding data availability:

R scripts and outputs are available here: https://osf.io/h5tvu/.

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
