# Peer review of "Problems in using p-curve analysis and text-mining to detect rate of p-hacking and evidential value"

_PeerJ, doi:10.7717/peerj.1715_

## Round 0.1 · original submission · Major Revisions

All the reviewers found the topic important and the modeling technique interesting as it offers new insights regarding the recently published study on the problem of p-value hacking. However, the reviewers have raised concerns about the details of the simulations as well as the presentation of the results. I would be happy to re-consider the manuscript if the major concerns are addressed adequately.

·

Basic reporting

The authors present simulations and re-analyses of data collected by Head et al (2014) to counter the argument that Head et al were able to quantify the amount of p-hacking. This is important – the paper by Head et al has received a lot of attention, but the analysis and conclusions are not very accurate. The commentary is selective, and focusses on only a few of the challenges of drawing conclusions from large numbers of p-values. The article addresses many concerns I have raised in response to a similar paper by De Winter & Dodou (2014), see Lakens, 2015.

The paper focusses on correlations between dependent variables. The authors could spend some more time on what they focus on, perhaps by giving an example. I believe they study situations where one collects either a set of uncorrelated DV’s, or a set of correlated DV’s. Many people have suggested that in psychology, almost everything is correlated with everything (e.g., in personality research, by Meehl) and going into some detail about this might set the stage regarding the relevance of these simulations. I believe Simonsohn et al (2011) provide a similar situation, but limit themselves to 2 DV’s – thus, the authors could specify they build on this, but extend it.

Evidence for publication bias has been around for a while (e.g., Greenwald, 1975). Please cite some older work to show this is not a recent problem.

Experimental design

For publication purposes, simulations based on only 10000 variables is not a lot. This is fine for quick tests, but if the article is going to appear in print, running 100.000 or 1.000.000 simulations to get highly accurate numbers is preferable (it is quite common to keep a computer running for 2 weeks if needed). For example, Figure 2 gives the idea of variability, while it should consist of straight lines. The spurious result in line 191 is similarly a distracting consequence of too few simulations.
Relatedly, the significance test related to “Note that the difference between frequency bins from .04-.045 vs .045-.05 was significant only for one simulation – the zero correlation case with medium power.” is complete nonsense. You are simulating data but could also describe the mathematical function that gives rise to this distribution. In other words, you can just say whether bins differ or not, mathematically. If they do, they always are ‘statistically different’ as long as you run the simulation with 1.000.000.000.000.000.000.000 simulations. You could make a comment about the power needed to achieve a significant result – but testing for it is meaningless.

Validity of the findings

Figure 1 needs to be improved – the uniform distribution does not end at 0.01 or 0.10, and the alternative distribution depends on power, which should be specified – as you note, most researchers don’t understand statistics very well, so it’s our responsibility to educate them.

The script has been annotated, but not yet for other users – it seems the comments are intended for the programmer, and make sense for the programmer, but it could be extended in a bit more detail. I found it difficult to understand the code, mainly due to the sometimes vague descriptions.

I tried to figure out myself why the p-values below 0.05 decrease, when p-values are correlated. This does not sound right – the percentage of type 1 errors should always be the alpha level if there is no Type 1 error inflation (note that plotting frequency on the y-axis in Figure 2 is not nearly as informative as plotting the percentage). Since 5% Type 1 error rates under the null is a mathematical reality, and p-hacking is supposed to inflate the Type 1 error rate, there is something unclear about these data. I would like the authors to examine this in more detail, or explain what I am missing.

Many tables in the main text are annotated for the authors, but not for the readers. In Table 1, not all columns are explained, but because they are included, I assume the authors want to communicate the information they contain. This is not yet successful. For example, I would expect the P.true.effect would be explained. The column names need periods (e.g., 0.04-0.045 instead of 04-045. The column ‘phack’ is again a significance test on simulated data, where a description (e.g., these numbers will be the same) or a difference in percentage (e.g., in the long run, there will only be 0.01% more p-values in bin X than Y) is more appropriate.

Additional comments

In lines 365-366, the authors comment on how “Our point, however, is that the p-hacking bump can be used to confirm that some p- hacking exists, but not to quantify the extent or impact of p-hacking.” That is fair enough, but the question is ‘what then’? We know that publication bias exists – and then we correct the effect size estimate. What is the use of concluding p-hacking is present?

One problem in p-curve analyses is heterogeneity. This is typically present (and definitely present in such a large set as Head et al) and it means there are different effect sizes, and different levels of power. Combinations can lead to surprising patterns of p-values, depending on their frequencies.

The authors state ‘To interpret a p-curve we need to have a model of the type of p-hacking, know whether the p-values were testing a specific hypothesis, and to be confident that if any p-values are excluded, the effect on the p-curve is random rather than systematic. Actually, the model you need is more complicated than this, and is explained in detail in Lakens, 2015 (also published in PeerJ) which you might want to cite (or you should extend the description of the model).

The authors state a problem, but not a solution to the problem. One solution would be for all future meta-analyses to share all meta-analytic data, including the data required for p-curve analyses. We recommend this in Lakens, Hilgard, & Staaks, in press. After some years, we would get a very accurate and reliable overview of how bad p-hacking is. I agree with the authors that the approach by Head et al does not tell us much, and it is good to correct the misinterpretations of their paper, but it would be good to end on a more constructive note, and highlight what can be done.



References:

Greenwald, A. G., (1975). Consequences of prejudice against the null. Psychological Bulletin, 82, 1-20.
Lakens, D. (2015). On the challenges of drawing conclusions from p-values just below 0.05. PeerJ.
Lakens, D., Hilgard, J., & Staaks, J. (in press). On the reproducibility of meta-analyses: Six practical recommendations. BMC Psychology.

·

Basic reporting

Please see general comments to the author

Experimental design

Please see general comments to the author

Validity of the findings

Please see general comments to the author

Additional comments

We like some parts of this critique, but not others. It’s true that our text mining analysis cannot provide much information about the number of papers with evidential value, or even the mean strength of evidential value, because it likely recovers some “bad” p-values (e.g. ‘boring’ tests checked model fit, etc) as well as missing lots of “good” ones (e.g. all p-values presented with a less-than sign). It’s also neat to present a model showing that ‘ghost variable p-hacking’ can influence the shape of the p-curve differently to other forms of p-hacking.

However we have a number of issues with the manuscript as it is. In general terms, it seems quite vague and unfocused. In many places, its criticisms apply to any and all studies that collect p-values and use them to measure p-hacking or evidential value, and thus it seems like a missed opportunity to frame this paper as a critique of ours, rather than a reappraisal of the p-curve approach in general. The text spends plenty of time simply reciting our methodology or repeating analyses that are already in our paper with very minor modifications, which unsurprisingly get qualitatively identical results. We recommend culling all the analyses that do not yield new insights, or point to concrete problems with our paper. We also think an opportunity has been missed with the “ghosthack” model. It is very flexible, yet the authors only investigate a small part of the parameter space. (See comments by Luke Holman on the pre-print http://tandf.figshare.com/authors/Luke_Holman/778667).

Another big issue is that we believe the authors present no new information that challenges our paper’s main conclusion, namely that we find evidence that p-hacking is occurring wherever we look. The authors appear to agree with this conclusion in several places, so we would appreciate it if they made it clear in the Abstract and Title of their paper that our main qualitative result is not in question. They do challenge our conclusion that the text mining indicates evidential value in the literature, but this conclusion is backed up by the meta-analysis work. Moreover, the authors collected some new data by manually collecting p-values from 30 of our text mined papers, and they could use these data to test if our conclusions change if the ‘bad’ p-values are excluded – puzzlingly they don’t do so, and we would be interested to see such an analysis.

Finally, there appear to be mistakes in some of the new analyses, e.g. due to miscounting numbers in the Tables in our paper, which qualitatively change the conclusions of the new analyses when corrected.

Yours Sincerely,
Megan Head and Luke Holman

Detailed comments

Line 22-25: This sentence implies that we made this conclusion solely on the basis of our text-mined data, when actually this conclusion derived from both our text mined data and data from meta-analyses. The text mined data simply demonstrated that p-hacking is widespread. The meta-analysis data found strong right skew in the p-curve, suggesting that real effect sizes are generally large, and we are not aware of any models that suggest that p-hacking can produce ‘spurious’ right-skew. If anything, your ghost hack model seems to suggest that p-hacking would reduce the evidence for evidential value, because it adds extra data in the range 0.3-0.5, weakening the apparent right skew.

In a comment of your preprint manuscript we suggested that your paper would be stronger if you focused on presentation of your paper as a critique of the methods rather than a critique of our paper specifically. Our paper is one example among others that uses text-mined data to generate p-curves and assess p-hacking in the literature, albeit on a much larger scale. To this effect, in the background section of the abstract we suggest you talk more generally about the problem and then in the methods section of the abstract (line 29-30) perhaps state “to demonstrate the problems associated with constructing p-curves from text mined data we examine the data-set created by Head et al”

Line 36: You state “the results of Head et al are further compromised…”. Actually we don’t think that your result presented in the previous sentence compromises our results. Our text mined data showed the p-hacking was widespread. The result in your previous sentence suggests our method (and indeed, all p-curve analysis) cannot detect some forms of p-hacking, which would bolster rather than compromise our result.

Line 39-40: You write “There was no information on the statistical power of studies nor on the statistical test conducted.” This information is not needed for tests of evidential value or p-hacking, and so this sentence should be deleted. If you want to keep it, you need to explain why it is important to have this information or why it invalidates studies using text mining to detect p-hacking.

Line 41-42: This doesn’t seem to be a problem with our paper - we found a bump and so we concluded there was evidence for p-hacking in our dataset. Perhaps it would be better to state your conclusions more generally. For instance, the p-curve is not an ideal method for detecting p-hacking because it doesn’t detect all forms of p-hacking, while a bump presents evidence of p-hacking the lack of a bump does not necessarily mean there is no p-hacking.

Line 43-44: You say “we cannot treat a right-skewed p-curve as an indicator of the extent of evidential value”. This is true for text-mined data. However, previous papers (e.g. . Simonsohn, U., L. D. Nelson, and J. P. Simmons. 2014. p-curve and effect size: Correcting for publication bias using only significant results) suggest that the p-curve can be used to estimate average effect size. Of course this is most useful for examining data that all relate to a specific question (similar to that presented in our analyses of previously published meta-analyses). You thus need to be clearer here that you are referring to the issues you outline with our text-mined data.

Line 69-71: This definition is not quite right. Simonsohn et al 2014 states “Thus, rather than file-drawering entire studies, researchers may file-drawer merely the subsets of analyses that produce non-significant results. We refer to such behavior as p-hacking.” The difference between your definition (the publishing of subsets of data) and Simonsohns definition (the publishing of subsets of analyses), is quite different, and your definition doesn’t cover all forms of p-hacking.

Line 72 and elsewhere: It would be useful to clearly distinguish between response variables and predictor variables throughout. The inclusion or exclusion of either of these both constitute p-hacking and likely have different effects on the shape of the p-curve.

Line 122-124: Our conclusion that p-hacking is weak relative to true effects and thus unlikely to drastically alter scientific consensus came from our analysis of previously published meta-analyses. See our abstract “Here, we use text-mining to demonstrate that p-hacking is widespread throughout science. We then illustrate how one can test for p-hacking when performing a meta-analysis and show that, while p-hacking is probably common, its effect seems to be weak relative to the real effect sizes being measured. This result suggests that p-hacking probably does not drastically alter scientific consensuses drawn from meta-analyses.” Thus, our conclusions did not derive from the text mining alone, as you imply.

Line 125-127: Contrary to what is written here, your model shows that p-hacking with ghost variables can produce an excess of p-values near 0.05. The p-curves for the most part are left skewed. Although the excess of p-values may be subtle and undetectable using a test for p-hacking we think your investigation of this could be clearer (see below).

Line 130-133: “many of the p-values were not suitable for p-curve analysis and may have introduced systematic bias into the p-curve” This statement refers only to our tests for evidential value. As such you should revise it; perhaps something like “many of the p-values were not suitable for inclusion in analyses using the p-curve to estimate evidential value”.

Figure 2: Since we are interested in the shape of the expected distribution of p-values it would be good to have smooth curves. This can be achieved by running simulations with more iterations (see archived code by Luke Holman http://tandf.figshare.com/authors/Luke_Holman/778667 for ways to run larger numbers of iterations very quickly). Furthermore, in simulations conducted by Luke Holman, he showed that the number of ghost variables measured has large effects on the p-curve. It would be good to include results when varying this parameter – I am not sure if you mention how many ghost variables were used when generating Figure 2.

Lines 151-155 and 189-193: Rather than doing binomial tests on individual simulation runs, as you did, It would be more informative to get a precise estimate of the frequency of p-values in the two bins being compared (e.g. by simulating 10^5 or 10^6 p-values). These estimates would allow you to do power analysis to work out the probability of detecting a difference of the predicted magnitude, for any given sample size. See here for some advice: http://stats.stackexchange.com/questions/38439/power-analysis-for-binomial-data-when-the-null-hypothesis-is-that-p-0). Note that the probability of getting a significant result declines to a minimum of 0.05 (for a flat p-curve) – thus it is not surprising that you got that one spurious result mentioned on line 191.

For example, if the simulation predicted that 6% of the p-values should fall in the lower bin and 5% in the upper bin (i.e. a modest difference), then one can use power analysis to show we’d have 92% power if we could collect 10,000 p-values in total (we think!). But actually this is kind of a moot point – estimating the shape of the p-curve by simulation requires making assumptions about the distribution of true effect sizes, the true proportion of papers that p-hack, and the type of p-hacking used (among other things), all of which are unknowable. Thus, it isn’t particularly useful to investigate lots of models, or suggest (as you do) that your simulation demonstrates that p-hacking will or will not be easy to detect in the real world.

Lines 195-198: Why did you choose to alter the effect size for the variables within a run? We think it would be better for all 10 variables to have the same non-zero effect size, because surely ghost hacking does not depend on some variables having stronger true effects than others, and thus this modeling decision detracts from your message. If you choose to keep your analysis as is you need to give a good justification for this choice. Furthermore, it would be good to run this analysis both when the ghost variables are correlated (e.g. r=0.8) and when they are not (r=0). This would allow readers to assess your results over a range of values, and would be interesting, because we think that ghost variables are probably often strongly correlated. Incidentally, doing the analysis with r=0.8 will probably yield a bump near 0.05 – this was the case in Luke Holman’s re-analysis (http://tandf.figshare.com/authors/Luke_Holman/778667)

Line 228: Not only is Hartgerink’s approach “not ideal because it leads to loss of power”, but it also incorporates p-values reported to 2 decimal places. This means he has included datapoints that Head et al., Bishop and Thompson, and Hartgerink all agree probably suffer from serious biases that hinder their interpretation. Please revise to clarify the main problem with his approach.
Lines 230-232: This is not what we said: here is what we wrote: “For example, it seems probable that p values of 0.049 will very often be reported as p < 0.05 instead of p = 0.049 or p = 0.05, due to authors trying to hide the fact that 0.049 is ‘only just significant’. ”
Figure 3: It seems there is a missing spike at p = 0.05. Compare your figure with Hartgerink’s.

Lines 253-254: Here, you claim that the proportion of p-values falling in each bin differs between set A and set B, but you do not present any statistics to support this. For example, based on the data in Table 2, one can see that the p-hacking test for multidisciplinary studies went from being significant in set A to being non-significant in set B. The number p-values in each of the two bin went from (666,783) in set A to (575,631) in set B. A Chi-square test shows that this is not a significant change in proportion (X^2 = 0.78, p = 0.38), and using a binomial GLM, one can estimate the change in proportion from set A to set B as 0.069 (95% CIs: -0.08 - 0.22). Thus, your conclusion is not warranted – this appears to be true for all the results that changed between sets A and B.

Table 2 appears to use a different method for estimating p-hacking than we did. We noticed this because you found no evidence of p-hacking in psychology using our dataset (46 papers in both the upper and lower bins) while we found good evidence of p-hacking (29 in the lower bin, 50 in the upper bin). The chance that you got 46+46, while we got 29+50, due to random sampling of papers is low, assuming you used the same sampling method as us (because we sampled one random p-value per paper 1,000 times, then took the average).We recall from our discussions of your pre-print that you discovered a bug in our code on Dryad, which could explain this discrepancy, assuming you used the corrected code. Is that what happened? Another issue is that the sum of the numbers in columns 2 and 3 of your Table 2 do not match the corresponding numbers in our Table 1, and I think they should. For example, your table suggests there were 24 “Agricultural” papers while our table suggests there were 26.

Line 275: This sentence implies that HEA concluded p-hacking is rare, but we don’t think we did. For example in the Abstract we wrote “p-hacking is probably common”

Lines 292-338: In this section you highlight that our method for looking at evidential value is on one hand conservative because we exclude p<, so we are likely to exclude a lot of very small p-values. But that on the other hand, we often erroneously include p-values from tests of models or ‘obvious’ results that will inflate our evidential value estimate. Do these two effects simply cancel out? I suspect the first effect will be much stronger – you showed that we missed around half of the very small p-values due to them being reported as inequalities, and you also imply that the “junk” p-values constitute the minority. We agree our test for evidential value is very crude, but in the end it might actually be a fairly accurate assessment of the literature. Perhaps it would be worth actually looking at the p-curve for your 30 papers using properly collected p-values (and including the very small ones) to determine whether our analysis really is likely to produce a false positive in tests for evidential value? And also, remember that the meta-analysis data (which does not suffer the problems outlined here) found the same conclusion.

Line 347: You could add “which is consistent with p-hacking” to this sentence.

Lines 34-3497: You appear to have made a mistake tallying the number in our Table 3. You say that there were 27 values in the lower bin and 40 in the upper bin, but actually there are 25 and 40. As reported in our manuscript this is a significant difference (binomial test: p = 0.040), not non-significant as you report (NB we already presented a binomial GLM for this result in the original paper, and so you should remove this section or spell out your reasons for re-testing the same data).

350-351: “Taken at face value, then, the evaluation of meta-analyses suggests p-hacking is rare” – For the record this was not our conclusion. We wrote “our results indicate that studies on questions identified by researchers as important enough to warrant a meta-analysis tend to be p-hacked.” Thus, you are misrepresenting our data (see above comment) and our conclusion.
Line 357: It’s not clear what you mean when you say that p-values need to be “on the same scale”.

Lines 360-363: This seems to be a more rounded description of your model’s results than you presented earlier. In your results section, you focus on the lack of significant difference in p-values between the upper and lower bins used for detecting p-hacking, rather than on the fact that there can be strong overall left skew in the p-curve. It would be good to revise your results section to be more in line with what you say here.

Line 365: You could also cite the comment by Luke Holman on your pre-print, which showed that ghost variable p-hacking can cause a bump in the p-curve

365: “Our point, however, is that the p-hacking bump can be used to confirm that some p-hacking exists, but not to quantify the extent or impact of p-hacking” We agree, mostly. Our original paper’s title is thus inaccurate. However, this seems to be a criticism of all p-curve based papers, so (as in our earlier comments) we suggest that your paper is better framed as a critique of p-curve papers in general, rather than ours in particular.

Table 4: This table gives the impression that using p-values from abstracts suffers fewer problems than using those from results. While we agree some problems are alleviated, you should probably also highlight that there are other problems with using p-values from abstracts. In many fields, the inclusion of p- in abstracts is optional, and so is usually only done when results are very strong. This bias is likely to overwhelm any benefits of using p-values from abstracts for measuring evidential value. Perhaps you could add another row to your table and a 3rd superscript (or restructure the table) to give a more balanced account of the pros and cons of using p values from abstracts and results. Likewise, limited power is one reason for using text mining to detect p-hacking, so perhaps a 4th superscript to highlight this point would allow readers to weigh up the pros and cons of different methods for generating p-curves.

Lines 386-388: Simonsohn et al. (2014) actually conclude that p-curve probably works fine for non-continuous data (their Supplement 4).

Line 405: You have not raised substantial doubt about our text mining conclusion with respect to p-hacking (though we agree you raise doubt about the evidential value analyses – though these are not the main focus of our paper). You agree with us in several places that our analysis is correct to conclude that p-hacking is occurring.

Reviewer 3 ·

Basic reporting

--

Experimental design

--

Validity of the findings

--

Additional comments

This well-written article handles a very important topic and is, as far as I can judge, scientifically correct. It gives new insights regarding recently published studies on the problem of p-value hacking.

I have specific concerns which may be addressed to improve the manuscript.

1. The terms reproducible/reproducibility are somethat confusing since they are used in the literature to denote different things. In particular, they have been used to denote the availability of data and computer code to allow readers to reproduce the results of the analysis from a computational point of view "by mouse click" based on the same data. As far as I can understand, in the present paper it is used in place of "validation": it is about reproducing the results in other studies, i.e. based on other data. This should be clarified. By the way, computational reproducibility may be useful to decrease the temptation of p-value hacking, but only if all ghost variables are made publicly available.

2. The term p-value hacking should be rigorously defined at the very beginning of the paper.

3. The connection between p-value hacking and multiple testing and the resulting type I error increase could emphasized more clearly.

4. Some interesting literature connected to p-value hacking (the authors may choose to cite a few them):

- Stanley Young
http://www.ncbi.nlm.nih.gov/pmc/articles/PMC2660953/
- Simmons et al (2011)
http://pss.sagepub.com/content/22/11/1359
- Ioannidis et al (2005), with a nice illustration: "Give me
http://www.ncbi.nlm.nih.gov/pubmed/15705441

5. line 126: "does not lead" -> perhaps rather "does not always lead".

6. Simulations: It is not clear how the observations are allocated to one of two groups to obtain the desired effect size. Please specify the data generating process clearly.

7. line 250: "bootstrapping process": What do you mean exactly? As far as I can understand bootstrapping is not used in the analysis.

8. line 346: "Interestingly, this was entirely due to inclusion of misrepoted p-values". I do not really understand this sentence and I am not sure you can affirm that.

9. The paper by Jager and Leek appeared in 2014.

---

## Round 0.2 · Minor Revisions

The manuscript has improved significantly, and most of the concerns have been addressed properly. However, due to large rewrites, the manuscript becomes a little unfocused as one reviewer points out. Therefore, the manuscript needs to be rewritten substantially to improve the readability, taking into account both reviewers' new comments. I also suggest removing the simulation of 'Skewed or Discrete data', and focusing on ghost variables, since it can be considered to be another form of p-hacking if we use inappropriate statistical methods (e.g. t-test for skewed data) to achieve statistical significance. Thus the appearance of the left skew of p-curve under this simulation setting still indicates evidence of p-hacking. The reason I grant a minor revision is that the methods and findings seem appropriate in this revision, but the interpretation and wording needs to be revised.

·

Basic reporting

The manuscript is very clear and well structured. The figures are clear, although some (Table 1, Figure2) might better be placed in supplementary material.

It's excellent the materials are all shared on the Open Science Framework. The Ghostscript 1.5 code is well annotated, and can be used by others to reproduce or extend these findings.

Experimental design

The simulations are valid, with reasonable parameters

Validity of the findings

Good.

Additional comments

I think overall this manuscript has improved substantially. I only have minor comments.

Abstract: “The p-curve is a plot of the distribution of p-values below .05” > why just below 0.05? The analysis is often done on p-values above 0.05 as well, and although publication bias makes the model more complex, it is possible.

L 129 Questions have, however, been raised as to whether p-curves provide a sufficiently robust foundation for such conclusions > Indeed, see also Ulrich & Miller, 2015: http://www.ncbi.nlm.nih.gov/pubmed/26595841 which might be worthwhile to cite

L 152: toward use > towards the use? (I’m not a native speaker).

Please relate the outcome from the simulations on skewed data directly to footnote 4 in Simonsohn etal: “As shown in Supplemental Material 2, p-curve’s expected shape is robust to other distributional assumptions.” - It seems your conclusions are not in line with this statement. Please explain to the reader what the differences in simulations are, and which simulations are more likely to apply to performed p-curve analyses such as Head et al (2015).

I now more clearly understand the simulations of correlated data. The explanation is better, but I would like to see a clear statement that data from a correlated (or dependent) t-test is, in principle, uniformly distributed. You now say “ the underlying distribution of all p-values follows a uniform distribution” but I would prefer to read: “the p-value distribution for a dependent or correlated t-test is uniform”. This is just to prevent confusion.
Your simulation is very similar to the situation where data is analyzed multiple times as more participants are added, when the data after participants 1-20 is correlated with the data after participant 1-40. The correlation is between different dependent variables, and the skewness comes from the fact you choose the best p-value. This could be added. These simulations are perhaps more accurately described by ‘dependencies in the data’ than by ‘correlated data’ since the choice for the lowest p-value is a crucial factor, which is a dependency (the lowest p-value) but not a correlation.

Line 262: range > perhaps: absolute number of Type 1 errors?

The addition of the section on power in lines 315-320 is excellent, very interesting.

Lakens (2015) is not in the reference list:
Lakens, D. (2015). On the challenges of drawing conclusions from p-values just below 0.05. PeerJ, 3:e1142, doi:10.7717/peerj.1142

·

Basic reporting

see general comments

Experimental design

see general comments

Validity of the findings

see general comments

Additional comments

The manuscript is much improved but still needs work before we believe it will be acceptable for publication. This partly reflects its multiple drafts and large rewrites during revision. The authors say in the intro that their focus is on ghost hacking, but this new version spends a lot of time talking about the expected p-curve under the assumption that all scientists do their statistics wrong (or work exclusively with categorical/binary response variables). It also includes bits and pieces of earlier drafts, which pop up unexpectedly later in the paper. So, it still needs a clear structure, aims, or objectives. We believe the paper is readable for people that are familiar with the p-hacking literature, but may cause problems for readers who aren’t.

The authors have responded well to many of our criticisms, principally by deleting much of the original paper, and adding new material that we suggested (e.g. the power analysis). However we still have numerous issues with the paper despite providing 3 previous rounds of review (two pre-prints, and one formal review). Additionally, many of the comments we have on this version are things we have highlighted in previous drafts. We are not sure if the authors disagree with our misgivings, or simply missed them – this was often unclear from their reply letter.

In a couple of places, in the manuscript the authors refer to earlier preprint versions of the manuscript. We are unsure of the protocol surrounding this practice, but it seems inappropriate since the reason those points are not in the current version of the manuscript is because they did not pass the scrutiny of peer review.

In a couple of places, the claims do not seem to match the data presented. The authors imply that their alternative handling of “rounded” p-values, presented in a previous Preprint, contradicted the results of Head et al, when as we recall it found the same results. The new MS also implies that its results demonstrate that Head et al include p-value data with the problems highlighted in Figure 3, when actually this was not tested.


Regards,

Luke Holman and Megan Head


Regarding the original comments in the previous round of review, we wrote:

Line 365: You could also cite the comment by Luke Holman on your pre-print, which showed that ghost variable p-hacking can cause a bump in the p-curve

And you replied:

This relates to the point we now make earlier in response to Daniel Lakens about variability around the p-curve

Presumably you are referring to Lakens comment 1.8? However this is not the same point that we were making. Our point is that your model shows very clearly that ghosthacking does cause left skew and possibly a bump in the p-curve, provided the ghost variables are correlated. We believe that highly correlated ghost variables are more likely to be the norm than the exception, because most of the time when people measure multiple variables during an experiment, they will be facets of the same process. Thus, ghost hacking may not be as undetectable as you suggest here (and as we have said before, most other forms of p-hacking can be detected in the p-curve, and our analysis did detect left skew). For example, in a nutrition study one might measure overall mass, fat content, and some physiological variables like the various types of metabolic rate, then only report the ‘good’ ones – these variables will be correlated to various degrees, not uncorrelated.

In your Response to Reviewers letter, you also wrote “We are not aware of any evidence that p-values in Abstracts are more likely to be reported when very strong”. As we recall, there is (unsurprisingly) a large difference in the p-values from the Abstracts and the Results in our p-value dataset (the Abstract ones are more significant) – have a look if you’re interested. Also, here is a paper where John Ioannidis calls Abstract p-values “a highly distorted, highly select sample.” (see arguments and references therein). http://biostatistics.oxfordjournals.org/content/15/1/28.long#ref-8





Specific comments on the new manuscript:

Line 46-48: The mismatch between people’s understanding of the foundations of statistics, and computing power is unlikely to be the most important driver of p-hacking. It was very possible to p-hack and have ghost variables even when people did stats by hand (‘accidentally’ looking up the wrong row in the t value table!). Motivation to find “significant” results is likely to be a more important driver, although increased computer power may make researchers more efficient p-hackers! You might want to consider reworking this.

Line 71-72: You say “our focus here is on ghost variables”. But that seems to be only part of the new paper, which focuses at least as much on other problems with the p-curve approach (like the fact that specific kinds of tests are expected to produce a p-curve bump even without p-hacking), and also with reanalysing Head et al. As such the focus of your manuscript is unclear and the MS would benefit from clearer aims and a more traditional structure.

113-114: ‘Quite simple binomial tests” It seems that the test used to detect p-hacking from the p-curve is not important: there are other tests that use the p-curve to test for p-hacking. For example, Mariscampo & Lalande 2012 fitted an exponential function to the p-curve, and tested whether the residuals tend to be more positive just under 0.05 than in other places (we decided this test was potentially more error prone than the binomial tests we settled on). Your paper highlights problems with using the p-curve to detect p-hacking not the specific statistical test that is applied to the question.

Lines 126-127: You could add that this test for p-hacking is conservative (at least assuming that most of the p-values we recovered are from tests under which the null p-curve is flat), because evidential value will tend to swamp the effect of p-hacking and make it harder to see the difference between the near and far bins. Thus, detecting p-hacking in spite of this issue, as we did, is tentative evidence that p-hacking is pretty common.

Line 144-145 “We show that p-curves do not follow a uniform distribution when…” This implies that you are showing this for the first time, but the quotes you provide by Simonsohn, and by Gelman and O’Rourke illustrate this result was already known. It would be worth clearly stating how what you show extends or fits in with what has been shown before, or revising your statement of priority. You could also state that this is an assumption that many studies that use p-curve to look for p-hacking in the literature (including Head et al, but also many others) ignore.

Line 144: “correlated dependent variables” There are still plenty of places where the MS could be more clear. What you mean to say here is “We simulated data to test how p-hacking using ghost variables affects the shape of the p-curve. We considered situations where researchers measure multiple response variables that are totally uncorrelated, weakly correlated, or strongly correlated with each other, and then only report one of the significant ones”

Line 155-156: Reword this a bit. Power always increases with the amount of available data, so maybe say e.g. “it is of interest to determine how much data is needed to have reasonable power to detect p-hacking”

Line 174-181: It seems like you have changed the simulation’s methods – or maybe it was always like this? Either way it seems a little wrong. It seems the correct way to do it is the way LH did it in his ghosthack model. Basically you make group A by drawing X variables with n replicates each from a multivariate normal distribution with mean 0 and SD 1. If you want them to be correlated or uncorrelated, make the appropriate tweaks to the correlation argument in R’s random multivariate normal function. Then do the same for group B. But if you want there to be a real difference between A and B, change the mean from 0 to something else (like 0.5 if you want the true effect size Cohen’s d to be 0.5). The way it is now, you make the puzzling and unexplained choice to make only one of the X variables have a higher expected mean, while the others still have an effect size of zero. It’s also unnecessary to roll a random number and then add a fixed value to it – just specify a higher mean for the random number. And it’s not needed to explain this at such length in the Figure.

Figure 2: This explains the mechanics of your code, but doesn’t do a greater job of explaining the structure of the model. You should stress that the ‘simulated scientist’ picks a randomly chosen significant p-value, and discards the others, leaving only one in the paper. This is the crux of ghost hacking, and it’s not in the Figure. At the moment, your figure mostly illustrates that the group that has a higher effect size is more likely to produce significant results, which is not useful. You also never explain how you mixed normal and log-normal data together, as you hinted on line 201 – did you assume that half the ghost variables are log normal and half are normal? Or did you combine 50-50 of each data type to make all X of the ghost variables? You also never say what test you used for the 6-category discrete response data. These should go into the Figure and/or text.

Figure 3 and line 222: The blue line is basically flat – it is not “something resembling a bimodal distribution”. It would also be useful to plot only/additionally the p-curve range, since this is the one that matters to p-curve studies (again, it looks like you haven’t run enough simulations to get a good grasp of the shape of the figure). And finally, you should point out more strongly that this model makes the very strong assumption that every single study misapplies a t-test to non-normal data (or every single study uses categorical data as the response variable). It seems very likely that the real world situation is not this bad (at least in my field, it’s pretty rare to see t-tests on non-normal data, because people are not so bad at stats any more), and so this figure does not totally invalidate p-curve studies as much as a naïve reader might infer. Maybe add these thoughts on line 236 – summarise the simulation results, then spell out their meaning for real world p-curve studies.

Line 258: The slope is not all that counter-intuitive (this makes Table 1 pretty unnecessary). There is p-hacking, which pushes real null results into the 0-0.05 range – so we see a pile up near the significance boundary. The only thing that is counter-intuitive is the lack of left skew when the variables are uncorrelated, and I think you could spend a little time explaining that to the reader.

Line 272-273: We’re not sure it’s good practice/appropriate to cite non-peer-reviewed comments on one of multiple PrePrints.

Line 284-285: Appendix one presents counts not percentages do you mean Appendix 3? Your appendices need figure legends.

Line 290: The left skew in your model is caused by the p-hacking, not by the fact that the variables are correlated

Line 296-300: Actually it doesn’t need to be p-hacking from ghost variables. Any type of p-hacking that produces left skew (basically all of them – except ghost hacking with 100% uncorrelated variables, which is presumably rare in the real world) would do it.

Figure 5: How high is “a high proportion of p-hacked” ? Tell us what your simulations do throughout the paper. Also, most people would agree that the p-hacking makes very little difference in this figure (the dotted and dashed lines are almost on top of each other), and so people might conclude that Simonsohn’s ‘rule’ that you criticise on line 297 is actually pretty good. Also, the red and green lines that almost overlap are very hard to distinguish for colour-blind people.

Line 315 onwards: The power analysis that we suggested is an improvement over the previous version, though we have the same complaint as last time: it’s basically fruitless to simulate the p-curve and then make quantitative inferences from it, because you are forced to use a very simplified, unrealistic model. Here, you have to assume that ghost-hacking is the only type of p-hacking going on (other types cause stronger left skew, making the test look weaker than it is), that all studies use exactly 8 ghost variables which have a correlation coefficient of 0.8, and many more. You also neglect to mention that the vast majority of life science research does not have the >0.8 power – from your graph, one can get a higher-than-average amount of power (~0.6) with only 300 samples. We still worry that readers will interpret this section as “Ghost hacking can be detected using p-curve, but it’s a low powered test, therefore Head et al are wrong” – though really this analysis doesn’t convincingly challenge our results for many reasons we have already outlined in our 3 previous reviews of this manuscript.

Lines 347-355: As we have pointed out before, it doesn’t matter to exclude p values like p<0.0001, because these are not informative when testing for p-hacking, making the inclusion of this whole section a bit needless.

Line 355- 359: Actually as we recall, your re-analysis (which took the drastic step of omitting all papers that had a least one p-value reported to 2 decimal places) found a qualitatively identical conclusion to ours (not a different conclusion as you imply here). Either put the analysis in the main paper, or remove this reference to the Preprint.

Line 340-381: Up until here the manuscript is focused very much on the detection of p-hacking. Here the manuscript switches to talking about ambiguous p-values in text mined data. Many of which are not relevant to tests of p-hacking (but rather tests of evidential value). Much of this discussion now seems out of place. You would be better to focus on ambiguous p-values that fall within the range 0.04-0.05 and demonstrate how much of a problem these are for tests of p-hacking in the sample of papers you checked.

Line 382-386: You could give a better explanation of what double dipping is. Also, you don’t explain why is this a problem for our study. We used a single p-value per paper, and if people practice double dipping in order to coax out significant p values, then these are in fact data that we want to find, not junk data.

Line 387: “Some p-values” How many? You collected this data, so say how many it was out of the total number of p values you assessed. As we argued before, these p-values are unlikely to fall in the range 0.04-0.05 and thus are not a problem for our tests of p-hacking.

Lone 400: ‘The data in Head et al do not adhere to these requirements’. Actually you did not measure whether the p-values you found in those 30 papers adhered to the ‘flat under the null’ assumption, so you are making a guess and misleadingly implying it is a result of your study. It is true that many/most of the p-values we gathered do not test the hypothesis of interest, in the Results section (though it’s probably ok in the Abstracts). But as we have explained repeatedly in previous reviews, this cannot create a ‘false positive’ in our test for p-hacking – it will actually lead to false negatives (since the primary results are not expected to be p-hacked). Since we did find ubiquitous evidence of p-hacking, we clearly did not experience a false negative due to this deviation from Simonsohn’s list.

Line 413-420: Here, you ignore the fact that we DID find evidence of p-hacking in the Abstracts. The text makes it sound like we did not even collect these data. This undermines your previous point about how our study includes lots of junk p values.

Line 429-430: “whether dependent variables were correlated” add “in the case that p-hacking was performed using ghost variables”. This assumption is not needed if p-hacking works in all the other ways (e.g. those proposed by Simonsohn).

Line 433: “amount of p-hacking” Has anyone ever claimed that the p-curve could be used this way? I’m pretty sure Head et al did not, and we don’t think Simonsohn does either. By the way when we said “extent of p-hacking” in the title, we had in mind the presence/absence across different fields, not the frequency within a field – though we see now this is ambiguous.

Line 436: “We have shown that…” see comment above regarding lines 144-145

Line 436-441: Yes this is an issue, but a key question that you never address is the frequency with which researchers misapply parametric tests to skewed data, or the frequency with which categorical tests appeared in our text mined dataset. You could easily test the latter with your sample of 30 papers. It seems unlikely that all, or even a majority, of papers misapply their tests as in your simulation, so we are sceptical that this seriously undermines Head et al’s conclusions.

Line 442: “usually” you have no data on whether ghost variables tend to be correlated or not, and we believe the correlated scenario is more plausible. Perhaps soften this section to something like “we have shown that some specific forms of p-hacking, such as ghost hacking when variables are weakly correlated, produce little or no left skew, and thus may go undetected by p-curve studies. Therefore, one should bear in mind that these tests are conservative”

Line 449: “the bump could have resulted for other reasons [than p-hacking]”. What are these other reasons? You don’t seem to give any. The only one you list, i.e. that researchers tweak their p-values and give the wrong one, for which we found evidence using recalculation, is a blatant form of p-hacking, so the sentence seems illogical. Also, the word “dangerous” seems a bit strong with regards to drawing conclusions from the presence of a bump. You have argued convincingly that bumps can result under a no-hacking scenario if most or all tests used incorrect stats, but this seems unlikely to occur in the real world. You’ve raised an important doubt, but we suspect you agree with Head et al that frequent p-hacking was likely a major reason for the bump.

Line 454-459: As we have said in previous reviews, you need to provide evidence that people actually make this mistake. Head et al do not, and we don’t know of any papers that do. So, it comes across as a swipe at Head et al, which is not justified. If you disagree, provide some examples/quotes of what previous work has said that you think is wrong – without it, we feel that you are making straw men.

Table 2: Change left column title to “not detected by p-curve analysis”. Second column title to “right skew in the p-curve”. The first, second and third points in the left column are essentially the same thing – the first one is a reason why there are few p values in the useful range, and the third one is a reason to exclude some p-values or papers, and thus to have low power. We would have thought problems associated with double dipping would also be alleviated by using abstracts of meta-analyses.

Acknowledgements: You might want to mention that we (LH and MH) have provided extensive input on the manuscript.

---

## Round 0.3 · Minor Revisions

Please consider the final (very minor) edits requested by the reviewer. After that, it is ready for publication.

·

Basic reporting

OK

Experimental design

OK

Validity of the findings

OK

Additional comments

I have only a few minor comments, and think this article is an interesting contribution to the literature, as it is..

Line 90: It is then likely that the result will be irreproducible. > It is then MORE likely

Line 260: this illustrates the difficulty of interpreting p-curves in real-life situations where parameters such as proportion of p-hacked studies, sample size and number and correlation of dependent variables are not known. > Please be more specific. What is the difficulty of interpreting p-curves in real life? The authors argue against Simonsohn in the next few lines, and you are correct in refuting the idea that all combinations of p-curves for which at least some effect exists produce right-skewed p-curves. However, when a p-curve indicates that a set of studies lacks evidential value, can we interpret it as problematic, always? I ask because recently I have seen people are p-curve is not always reliable, but what I think the take-home message is is that p-curve, like all meta-analytic techniques that check for bias, can tell you when something is problematic. However, when they say things look good, things might still be bad. Is that a fair description? Should people take away that p-curves are always difficult to interpret in real life situations, or only when they indicate the data might be fine, but there is always reason for concern when p-curve shows a lack of evidential value?

Line 357: This is a form of p-hackingthat > ADD SPACE

Line 431: “it can be difficult to interpret the absence of a p-hacking 'bump'” – it might make sense to note how almost all meta-analytic techniques that try to correct for bias work like this. E.g., the trim and fill method can show a problem, but not the lack of a problem, because it, like all tests, is based on certain assumptions. If these assumptions are not met, the results are difficult to interpret.

---

## Round 0.4 · accepted · Accept

The manuscript is now suitable for publication.